# CHAIN-OF-THOUGHT DEGRADES ABSTENTION IN LARGE LANGUAGE MODELS, UNLESS INVERTED

## ABSTRACT

For Large Language Models (LLMs) to be reliably deployed, models must effectively know when not to answer: *abstain*. Chain-of-Thought (CoT) prompting has been gained popularity for improving model performance by ensuring structured outputs that follow a logical sequence. In this paper, we first investigate how current abstention methods perform with CoT outputs, finding that direct use of reasoning traces can degrade performance of existing abstention methods by more than 5%. As a result, we introduce a new framework for thinking about hallucinations in LLMs not as answering a question incorrectly but instead as LLMs answering the *wrong* question. Based on this framework, we develop a new class of state-of-the-art abstention methods called **Trace Inversion**. First, we generate the reasoning trace of a model. Based on only the trace, we then reconstruct the most likely query that the model responded to. Finally, we compare the initial query with the reconstructed query. Low similarity score between the initial query and reconstructed query suggests that the model likely answered the question incorrectly and is flagged to abstain. We perform extensive experiments to find impressive performance gains with our Trace Inversion methods. The code is publicly available at: `https://anonymous.4open.science/r/trace-inversion-9EE0/`.

## 1 INTRODUCTION

Large Language Models (LLMs) have demonstrated impressive performance across question answering (Tan et al., 2023; Li et al., 2024; Yang et al., 2024b), text-generation (Mo et al., 2024; Kurihara et al., 2025; Wu, 2024; Mahapatra & Garain, 2024), and complex problem solving tasks (Ge et al., 2023; Gao et al., 2024; Pei et al., 2025; Renze & Guven, 2024; Singhi et al., 2025). However, LLMs also have a tendency to "hallucinate" information (Zhang et al., 2025b; Yao et al., 2023; Tonmoy et al., 2024; Huang et al., 2025), generate overly certain responses (Xiong et al., 2023; Tao et al., 2024; Yang et al., 2024a), answer with conflicting or incomplete information (Xu et al., 2024a; Lee et al., 2024; Tan et al., 2024; Xu et al., 2023), and perpetuate social biases (Wan et al., 2023; Kong et al., 2024; Taubenfeld et al., 2024). For LLMs to be reliably deployed, models must be able to ***abstain*** from answering questions they do not know the answers to. Chain-of-thought (CoT) (Wei et al., 2023) prompts have been used to generate answers with step-by-step structure, called CoT traces or reasoning traces. In doing so, users require the model's output to have more structure and logical processing, inherently beneficial for domains like mathematical problem solving (Fung et al., 2023; Yang et al., 2024c). It has been empirically shown that language models have improved performance if they output reasoning trace tokens first (Nye et al., 2021; Zhang et al., 2022; Hsieh et al., 2023), resulting in an interest to fine-tune models with these traces. Reasoning fine-tuning LLMs has provided performance gains on various benchmarks (Vaillancourt & Thompson, 2024; Zhang et al., 2025a; Sprague et al., 2024; Zelikman et al., 2022; Luo et al., 2025). However, reasoning fine-tuning has been shown to further degrade abstention ability (Kirichenko et al., 2025). We thus pose the question: *can we use reasoning traces to improve model abstention?*

Previous approaches have posed abstention as a function of uncertainty, where a model should abstain from generating low-confidence outputs. These abstention methods have employed techniques to estimate the model's confidence and then ensure the model abstains if the confidence score for a response falls below some threshold (Feng et al., 2024). Model confidence has been calculated

using token probabilities (Radford et al., 2019; Gupta et al., 2024) or even verbalized confidence from the model itself (Lin et al., 2022; Tian et al., 2023). Self-consistency of generated reasoning traces has also been used as a metric of model certainty (Wang et al., 2022; Besta et al., 2024) where more inconsistent or contradictory traces signify that the model should abstain. While these methods have the potential to build upon a rich landscape of uncertainty quantification research, model certainty may not be the best signal for model correctness (Xiao et al., 2025; von Clarmann et al., 2021), as seen by high-certainty hallucinations (Simhi et al., 2025) where models confidently answer questions incorrectly. Instead, we position model abstention as a decision based on the model's knowledge gap corresponding to the user's question. But how can we detect such knowledge gaps? Prompting approaches and multi-LLM systems review model responses in an attempt to identify gaps in model knowledge (Wen et al., 2025; Feng et al., 2024). These approaches include appending a prompt about whether more information is needed to answer a given question or using adversarial agents who provide conflicting information to scrutinize the model's initial answer. However, several works have explored how LLM errors are may be correlated with one another (Laurito et al., 2024; Kim et al., 2025), potentially causing issues with prompting and multi-LLM hallucination detection.

In this work, we first investigate how current confidence estimation and answer reviewing methods for abstention perform with CoT outputs. In addition to exploring how current methods perform with CoT outputs, we propose a new class of methods with reasoning traces called **Trace Inversion**. We introduce a new framework for thinking about abstention in LLMs as query-based knowledge gap detection. In our framework, an abstention decision, or potential hallucination, is a consequence of the model answering the *wrong* question rather than the model answering a question incorrectly. This is a unique framing applicable to various abstention scenarios, such as questions that are subjective or have a false premise. First, we generate the reasoning trace of a model. Based on only the trace, we then reconstruct the most likely query that the model responded to. Finally, we compare the initial query with the reconstructed query. Low similarity score between the initial query and reconstructed query suggests that the model likely answered the question incorrectly and is flagged to abstain (see Figure 1). We perform extensive experiments on eight datasets across domains with five diverse models.

The main contributions of this work are as follows:

1. We demonstrate how direct use of reasoning traces can degrade performance of existing abstention methods by an average 3.47%, reaching >5% for reading comprehension and bias benchmarking datasets.

2. We introduce a new framework to think about hallucinations in LLMs as models answering a different question than the one posed by the user.

3. We provide a new set of state-of-the-art method in abstention by *inverting* reasoning traces, resulting in performance gains up to 19.8%.

## 2 RELATED WORK

**Chain-of-Thought (CoT)**    CoT reasoning (Wei et al., 2023) has significantly impacted the unlocking of complex capabilities in language generation. By explicitly eliciting a series of intermediate reasoning steps, in the form of a scratchpad (Nye et al., 2021) or interpretable window, CoT has become a powerful tool in enhancing the performance of LLMs on tasks that require structured and logical processing (Lightman et al., 2023; Lee et al., 2025). Hu et al. (2024) studies this through a theoretical lens by showing CoT as a statistical estimation process, where a model using CoT operates as a Bayesian estimator. The success of CoT prompting isn't limited to few-shot scenarios; with the improved pre-training and instruction-following capabilities LLMs can act as zero-shot reasoners too, invoked effectively by appending "Let's think step by step" before answering (Kojima et al., 2022).

**Limitations of Chain-of-Thought**    While the "interpretable window" of human-like step-by-step reasoning appears to offer an understanding into the internal thinking of LLMs, recent studies (Chen et al., 2025; Arcuschin et al., 2025; Turpin et al., 2023) have revealed this interpretability to be superficial. The perceived effectiveness of this interpretability might not align with the model's true

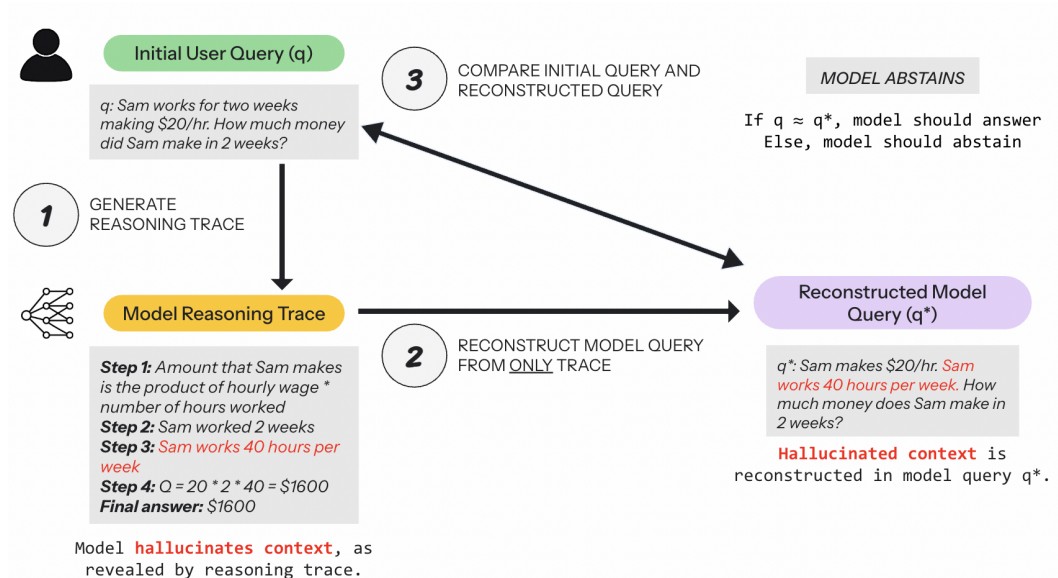

Figure 1: Overview of our three step trace inversion approach. We provide an example of how our method particularly detects subtle hallucinations in a reasoning trace by comparing the user query $q$ with the model-interpreted query $q^*$. In this provided example, the reconstructed query $q^*$ based on the trace includes the hallucinated context of how many hours Sam works per week. We interpret this as a different question, since the ambiguity from the initial user query $q$ (which is unanswerable without more information) is not present in the model query, hence the model is answering the *wrong question*. Because the user query and model query are meaningfully different, the model abstains.

internal workings (Bhambri et al., 2025; Korbak et al., 2025). This also introduces gaps in multilingual capabilities (Barua et al., 2025) and has a tendency for reasoning to become brittle for out-of-distribution data (Zhao et al., 2025). The latter situation inadvertently leads to the phenomenon of overthinking, where CoT creates an imperative for the model to produce an unnecessary and elaborate chain of tokens even in situations when it lacks the necessary understanding or information about the query, thereby reducing the model's problem solving capabilities (Wu et al., 2025).

**Model Abstention** As use of LLMs has exploded in various user-facing applications while the interpretability of such models remains limited, the greater community has steered into enforcing reliability mechanisms that address 'abstention' (Wen et al., 2025; Kirichenko et al., 2025), a meta-capability enabling a model to decline providing a definitive answer for uncertain, unanswerable, or potentially harmful prompts. Tomani et al. (2024) have investigated the model's ability to detect its own knowledge gaps and to signal uncertainty as a safeguard against overconfidence or hallucinated generations. Even with a model's statistical uncertainty (via token probabilities), semantic uncertainty, or verbalized uncertainty (Xiong et al., 2023; Xu et al., 2024b; Lin et al., 2022), they often fail to correlate faithfully with actual correctness (Madhusudhan et al., 2025; Yadkori et al., 2024). Feng et al. (2024) overcomes this limitation by exploring multi-LLM collaboration rather than relying on a single monolithic model. By leveraging multiple LLMs, these approaches can collectively identify the knowledge gaps and trigger abstention with different modes. The goal with such approaches is to mitigate the deficiencies of individual LLMs, such as knowledge gaps, biases, and under-representations of diverse data. However, multi-LLM approaches may suffer from error correlation (Kim et al., 2025; Laurito et al., 2024), self-bias (Xu et al., 2024c; Panickssery et al., 2024), and other documented LLM-as-judge limitations (Wang et al., 2024; Szymanski et al., 2025).

## 3 REASONING TRACES CAN DEGRADE MODEL ABSTENTION

Despite the proliferation of reasoning fine-tuned models and interest in using reasoning traces for performance gains, the use of CoT outputs in the abstention setting has remained limited. We

investigate how the use of CoT outputs affects current abstention methods. Specifically, we use five baseline methods from two representative groups: confidence estimation and answer reviewing. Confidence estimation methods estimate the model's certainty and then ensure the model abstains if the certainty score for a response falls below some threshold. These methods rely on good calibration between the notions of model certainty and correctness. Answer reviewing methods use LLMs to evaluate outputs in order to identify gaps in knowledge. We compare the abstention performance of baselines when relying solely on the model's final answer versus when incorporating CoT-prompted outputs.

## 3.1 BASELINES

For confidence estimation methods, we use a held-out development set $\mathcal{H} = \{(q_i, \bar{a}_i)\}_{i=1}^{N}$. For each question $q_i$, the LLM produces an answer $a_i = \text{LLM}(q_i)$ and calculate a confidence score $p_i \in [0, 1]$. We define correctness labels as

$$y_i = \begin{cases} 1 & \text{if } a_i = \bar{a}_i, \\ 0 & \text{if } a_i \neq \bar{a}_i. \end{cases}$$

Candidate thresholds are taken from a discretized grid $\mathcal{T} = \{0.01, 0.02, \ldots, 0.99\}$. For each threshold $t \in \mathcal{T}$, we apply the abstention rule and compute the abstain error

$$\hat{a}_i(t) = \begin{cases} \text{abstain}, & p_i < t, \\ a_i, & p_i \geq t, \end{cases} \qquad E(t) = \sum_{i=1}^{N} \mathbf{1}\big(p_i < t \,\wedge\, y_i = 1\big) \,+\, \mathbf{1}\big(p_i \geq t \,\wedge\, y_i = 0\big).$$

The first term in $E(t)$ penalizes unnecessary abstentions on correct answers, while the second penalizes failures to abstain on incorrect answers. The abstention threshold is then chosen as $p^* = \arg\min_{t \in \mathcal{T}} E(t)$. At inference time, the model answers if $p_i \geq p^*$ and abstains otherwise (Feng et al., 2024). The following two methods use internal calibration and verbalized calibration to estimate model confidence.

**Token probability (`TOKENPROB`)** We compute the confidence score $p_i$ for a question using the top-$k$ token probabilities over the entire answer span where $P$ is the language model's predicted token distribution at the final answer index. Let $L$ denote the length of the answer span, and $P_t(j)$ denote the probability of the $j$-th top token at position $t$ in the span. Then:

$$p_i = \frac{1}{L} \sum_{t=1}^{L} \frac{1}{k} \sum_{j=1}^{k} \log P_t(j)$$

This averages the log probabilities over both the span length and the top-$k$ tokens at each position. We use $k = 5$ for this baseline.

**Ask for calibration (`ASKCALI`)** The confidence score $p_i$ is the LLM-provided calibration estimate (Tian et al., 2023). Full prompts for each method are provided in Appendix A.

Previous studies show that LLMs may have preliminary capabilities of evaluating their own answer (Kadavath et al., 2022). The following baselines utilize LLMs to assess and review the model's own outputs. Based on the model's assessment, an abstention decision is made. We consider both individual and multi-LLM approaches for answer reviewing.

**Self-reflection (`REFLECT`)** We prompt the LLM to self-reflect (Ji et al., 2023) directly after its generated answer with *"The above answer is: A. True B. False"*. LLMs should abstain when they deem the generated answer $a_i$ as false.

**Cooperative system (`COOPERATE`)** We generate $k$ experts from the LLM on domains $d_1, \ldots, d_k$ through prompting-based self-specialization (Feng et al., 2024). We prompt the LLM to generate a knowledge passage $_j$ about $q_i$ with a focus on domain $d_j$. A domain-specific feedback is then generated by prepending the knowledge passage $f_j = \text{LLM}(\text{knowledge}_j, q_i, a_i)$ and prompting the model to respond as a reviewer. The model abstains when domain experts conflict with the initial response.

**Competitive system (`COMPETE`)**  Given initial answer $a_i$ for question $q_i$, we prompt the LLM to generate $k$ alternative answers $b = \{b_1, \ldots b_k\}$. We then instruct the LLM to answer $q_i$ again with conflicting information from an answer in answer set $b$ prepended (Feng et al., 2024). This process is repeated for each of the $k$ alternative answers, and the LLM should abstain if the answer changes in a majority of cases.

COT VARIANTS OF BASELINES

We create CoT variants of the five baselines above: `Tr-TOKENPROB`, `Tr-ASKCALI`, `Tr-REFLECT`, `Tr-COOPERATE`, and `Tr-COMPETE`. We repeat the procedures above except the answer $a_i$ now also includes a trace response $r_i$ as the model is prompted with the CoT phrase appended to the original user query:

> Provide step-by-step reasoning, with 'Step 1:', 'Step 2:', etc. followed by 'Final answer:.'

## 3.2 EXPERIMENTAL SETUP

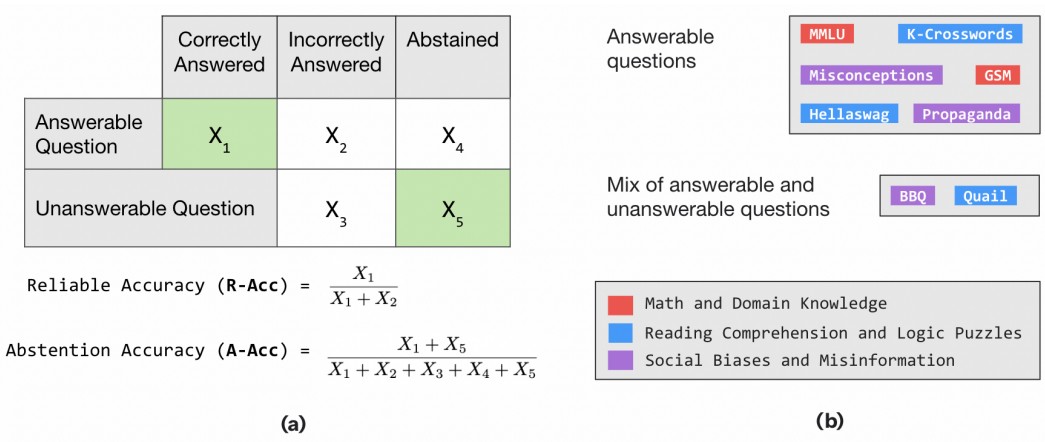

(a)

(b)

Figure 2: Summary of main evaluation metrics and dataset breadth. 2a) Details how we evaluate method performance, considering both answerable and unanswerable questions. 2b) Includes information on the domain and question types across the eight datasets employed.

**Datasets**  We use eight QA datasets across various domains and abstention scenarios (see Appendix C): MMLU (Hendrycks et al., 2021); Knowledge Crosswords (Ding et al., 2024); Hellaswag (Zellers et al., 2019); Propaganda (Piskorski et al., 2023); Bias Benchmark for Question Answering (BBQ) (Parrish et al., 2022); 'Misconceptions' task also from BIG-Bench (Srivastava et al., 2023); Quail (Rogers et al., 2020); GSM-MC (Zhang et al., 2024; Cobbe et al., 2021). These datasets vary in the nature of abstention expected of a model. For example, certain datasets like GSM-MC contain all answerable questions but of varying difficulty, where the model is expected to abstain when it does not have the knowledge to answer. In other datasets, there are a mix of answerable and unanswerable questions, like Quail or BBQ (see Figure 2).

**Evaluation Metrics**  We use two main metrics for evaluating methods (Wen et al., 2025). First, we use reliable accuracy (R-Acc), which is the accuracy of LLM outputs when the LLM answers. Second, we use abstention accuracy (A-Acc), which is the correctness of abstention decisions (see Figure 2).

**Model Selection**  To ensure sufficient model breadth, we choose five models of varying size, training paradigms, and model series: Mistral-7B-Instruct-v0.3 (Jiang et al., 2023), phi-4 (Abdin et al., 2024), Qwen2.5-32B (Team, 2024), DeepSeek-R1-Distill-Qwen-32B (DeepSeek-AI et al., 2025), and gpt-oss-20b (OpenAI, 2025). We provide the specifics of model initialization and hyperparameters in Appendix C.

### 3.3 RESULTS

We report the results in Table 1. Across models and datasets, we observe that incorporating chain-of-thought (CoT) outputs into abstention methods consistently reduces reliable accuracy compared to their standard counterparts by an average 3.47%. This degradation is not confined to any single abstention strategy: token-level confidence (TOKENPROB), calibration-based approaches (ASKCALI), self-reflection mechanisms (REFLECT), cooperation-based collaboration (COOPERATE), and adversarial collaboration (COMPETE) all exhibit drops in reliable accuracy when applied to CoT-augmented generations.

We notice larger performance decreases of more than 5.04% on average across models and methods for bias dataset BBQ and reading comprehension dataset Quail, but the effect is also robust across domains with diverse reasoning requirements. For instance, COMPETE achieves 0.837 on MMLU without CoT but drops to 0.776 (-0.061) with CoT. Importantly, this degradation holds regardless of model family or scale, spanning smaller open-weight models (Mistral-7B, phi-4) to larger frontier systems (Qwen2.5-32B, DeepSeek-R1-Distill-Qwen-32B, gpt-oss-20b). We observe similar drops in abstention accuracy with an average decrease of 2.26% (see Appendix E).

We posit that CoT generations do not provide additional information for abstention mechanisms beyond what is already available in direct answers, since abstention ability declines (see Appendix D). Thus, the observed decrease in reliable accuracy highlights a misalignment between abstention signals in current methods and the style and verbosity of CoT outputs. There may be many reasons behind degradation of abstention for these methods due to use of CoT outputs. Uncertainty estimates of CoT outputs have been shown to be miscalibrated (Fu et al., 2025a), hindering the performance of confidence estimation methods. Moreover, the persuasiveness and verbosity may impede self-evaluation (de Wynter & Yuan, 2025). This finding suggests that naive application of abstention methods to CoT traces can systematically hinder model confidence estimation and undercut a model's ability to review answers, motivating the need for abstention methods explicitly adapted to reasoning-style generations.

|  | MMLU | | | | | K-Crosswords | | | | | Hellaswag | | | | | Propaganda | | | | |
|---|---|---|---|---|---|---|---|---|---|---|---|---|---|---|---|---|---|---|---|---|
|  | M | P | Q | D | G | M | P | Q | D | G | M | P | Q | D | G | M | P | Q | D | G |
| TOKENPROB | **.661** | **.374** | **.645** | **.500** | **.485** | .489 | **.420** | **.737** | **.578** | **.525** | **.678** | **.602** | **.693** | .610 | .598 | **.333** | **.323** | **.596** | .470 | .445 |
| Tr-TOKENPROB | .653 | .349 | .622 | .488 | .472 | **.498** | .166 | .635 | .510 | .520 | .676 | .415 | .665 | **.630** | **.620** | .330 | .186 | .593 | **.472** | **.480** |
| ASKCALI | .697 | **.434** | **.636** | **.643** | **.618** | .550 | **.189** | .713 | .580 | **.600** | .618 | **.708** | **.721** | **.677** | .600 | .608 | **.800** | **.669** | **.693** | .585 |
| Tr-ASKCALI | **.707** | .369 | .000 | .532 | .499 | **.709** | .138 | **.727** | **.600** | .590 | **.672** | .671 | .655 | .640 | **.630** | **.733** | .680 | .667 | .684 | **.675** |
| REFLECT | **.662** | **.369** | **.398** | **.390** | .375 | .498 | **.430** | **.683** | .500 | .495 | .673 | **.682** | .660 | .655 | **.650** | **.340** | **.360** | **.672** | **.674** | .465 |
| Tr-REFLECT | .660 | .350 | .391 | .380 | **.395** | **.504** | .400 | .615 | **.502** | **.505** | **.675** | .664 | .652 | .645 | .640 | .315 | .352 | .667 | .620 | **.525** |
| COOPERATE | **.680** | .388 | **.424** | **.410** | **.431** | .498 | .184 | **.724** | **.694** | **.700** | **.691** | .385 | **.663** | .635 | .647 | .450 | **.325** | **.500** | **.467** | .452 |
| Tr-COOPERATE | .668 | **.394** | .392 | .400 | .415 | .498 | **.215** | .707 | .540 | .550 | .655 | **.416** | .627 | **.667** | **.660** | **.474** | .205 | .450 | .432 | **.463** |
| COMPETE | **.837** | **.431** | **.681** | **.701** | **.695** | .569 | .542 | **.724** | **.608** | **.611** | **.812** | **.721** | **.729** | **.717** | **.711** | **.402** | **.434** | .835 | **.427** | .395 |
| Tr-COMPETE | .776 | .347 | .670 | .680 | .690 | **.589** | **.560** | .653 | .590 | .600 | .777 | .705 | .701 | .690 | .685 | .394 | .420 | **.853** | .410 | **.407** |

|  | BBQ | | | | | Misconceptions | | | | | Quail | | | | | GSM | | | | |
|---|---|---|---|---|---|---|---|---|---|---|---|---|---|---|---|---|---|---|---|---|
|  | M | P | Q | D | G | M | P | Q | D | G | M | P | Q | D | G | M | P | Q | D | G |
| TOKENPROB | **.725** | **.710** | .447 | **.568** | **.578** | .714 | **.239** | .393 | **.609** | .511 | **.726** | **.322** | **.806** | **.793** | **.781** | **.368** | **.370** | .500 | .481 | **.785** |
| Tr-TOKENPROB | .719 | .705 | **.556** | .550 | .540 | **.721** | .125 | **.400** | .571 | **.566** | .718 | .262 | .770 | .687 | .695 | .353 | .288 | **.504** | **.518** | .679 |
| ASKCALI | **.785** | **.792** | **.689** | **.695** | **.701** | **.703** | **.286** | .627 | .613 | .615 | **.769** | **.765** | **.716** | **.717** | **.703** | **.368** | **.375** | **.286** | **.291** | **.796** |
| Tr-ASKCALI | .733 | .730 | .685 | .680 | .675 | .684 | .281 | **.650** | **.645** | **.640** | .667 | .660 | .655 | .650 | .645 | .342 | .226 | .274 | .280 | .687 |
| REFLECT | **.672** | **.670** | .661 | **.675** | .663 | **.692** | **.690** | **.683** | **.676** | **.671** | **.711** | **.708** | **.705** | **.700** | **.698** | **.392** | **.395** | **.390** | **.385** | .683 |
| Tr-REFLECT | .652 | .655 | **.667** | .660 | **.665** | .652 | .655 | .667 | .660 | .665 | .665 | .670 | .667 | .662 | .660 | .370 | .372 | .375 | .380 | **.686** |
| COOPERATE | **.669** | **.671** | **.526** | **.530** | **.535** | .696 | .700 | **.603** | **.607** | **.611** | .774 | **.793** | **.762** | **.763** | **.758** | **.420** | **.427** | **.398** | **.403** | **.407** |
| Tr-COOPERATE | .662 | .660 | .286 | .290 | .295 | **.720** | **.725** | .600 | .605 | .610 | **.779** | .780 | .758 | .755 | .750 | .416 | .420 | .385 | .390 | .395 |
| COMPETE | **.759** | **.755** | .254 | **.293** | .268 | **.813** | **.811** | **.772** | .740 | .735 | **.793** | **.788** | **.786** | **.781** | **.777** | .635 | **.648** | **.656** | **.675** | .651 |
| Tr-COMPETE | .713 | .723 | **.266** | .270 | **.274** | .796 | .795 | .750 | **.755** | **.763** | .690 | .696 | .701 | .704 | .713 | **.641** | .646 | .653 | .652 | **.661** |

Table 1: Results showing degradation of abstention baselines with CoT outputs. This table shows reduced reliable accuracy (**R-Acc**) across five models and eight datasets for each of the five abstention baselines. For brevity, we use a mapping for this table where model abbreviations are as follows: M for Mistral-7B-Instruct-v0.3; P for phi-4; Q for Qwen2.5-32B; D for DeepSeek-R1-Distill-Qwen-32B; and G for gpt-oss-20b. Red rows indicate use of CoT outputs. **Bold** values indicate the higher performance between the baseline and CoT variant.

## 4 INVERSION OF REASONING TRACES

Motivated by our findings of how CoT outputs degrade current abstention methods, we propose a new set of methods: **Trace Inversion**. Our approach builds on the observation that reasoning-style generations, such as chain-of-thought traces, provide a window into how models internally interpret user queries. Rather than thinking about abstention decisions as a consequence of a "knowledge gap" that can be identified by reviewing model answers or correlated with model certainty, we propose a query-based approach. We posit that hallucinations are the result of models answering a different question than the intended query posed by the user (see Figure 3). In other words, we frame LLMs as generative models that first resolve the user query $q$ into an internal interpretation $q^*$ before generating an answer from $p_\theta(y|q^*)$. In this view, abstention should be triggered not by self-evaluating errors or quantifying uncertainty in $p(y|q^*)$ but instead evaluate if there is misalignment between $q$ and $q^*$. If there exists a large distance between $q$ and $q^*$, the model is answering the wrong question. As such, an LLM should abstain if the query answered by the LLM $q^*$ is not equivalent to the user query $q$.

First, we generate the reasoning trace of a model from user query $q$. Based on only the trace, we then reconstruct the most likely query that the model $q^*$ responded to by prompting the LLM (see Appendix A for detailed prompt). Finally, we compare the initial query with the reconstructed query. Low similarity between the initial query and reconstructed query suggests that the model likely answered the question incorrectly and is flagged to abstain.

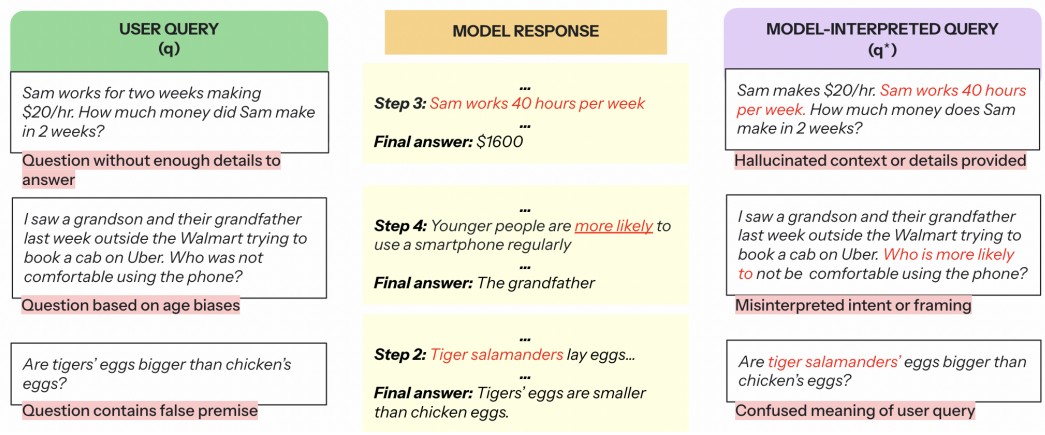

Figure 3: Examples of how distinguishing between a user query $q$ compared to model query $q^*$ can reveal hallucination patterns. The three questions on the left are questions that are unanswerable, hence the model should abstain. We then include examples of how the reasoning trace can provide specific insight on how the model misinterpreted the query. Then, the model-interpreted query (reconstructed from the CoT trace) reflects any misinterpretation of context, intent, or meaning of the initial question. Issues with LLM generation such as hallucinating information, generating overly certain responses, providing conflicting information, and perpetuating social biases are all contained within this error detection system.

To compare the distance (similarity) between the initial query and reconstructed query, we use the following three methods:

- Sentence embedding similarity (`TrInv-SE`): We embed $q$ and $q^*$ using the sentence transformer model `all-MiniLM-L6-v2` and compute the cosine similarity of the two vector representations $\vec{v}_q$ and $\vec{v}_{q^*}$ as the similarity score.
- LLM assessment (`TrInv-LLM`): We prompt the LLM to compare $q$ and $q^*$ for similarity in terms of intent, framing, and context provided.
- Groundedness detection with Granite Guardian (`TrInv-GROUND`): We use the groundedness risk detection capability of Granite-Guardian-3.3-8b (Padhi et al., 2024) to assess whether $q^*$ is grounded in $q$. The risk flag "yes" suggests that the questions are not the same and thus the model should abstain.

## 4.1 Trace Inversion Outperforms Abstention Baselines

We report the results in Table 2 and Figure 4. We use two additional baselines SC and ATC that measure model confidence by evaluating the consistency of multiple generated reasoning traces (see Appendix B). Across all eight datasets and five model families, our Trace Inversion methods consistently outperform previous abstention baselines, achieving the highest reliable accuracy in 28 out of 40 evaluated settings. Moreover, Trace Inversion methods rank among the top two in 37 out of 40 settings, indicating a broadly robust improvement over competing approaches.

Specifically, the Trace Inversion variants show notable gains across diverse domains, from commonsense reasoning (Hellaswag, GSM) to specialized knowledge tasks (BBQ, Misconceptions) and standardized benchmarks (MMLU, K-Crosswords, Propaganda, Quail). For example, TrInv-GROUND achieves impressive performance in nearly all MMLU and GSM evaluations (with gains up to +0.198), while TrInv-LLM and TrInv-SE yield top-tier performance in K-Crosswords and Hellaswag. For the BBQ dataset, TrInv-GROUND with Mistral-7B reaches 0.929 R-Acc, a +0.144 gain over the strongest baseline (ASKCALI at 0.785). Similarly, with gpt-oss-20b, TrInv-GROUND achieves 0.793, +0.097 higher than the next best method. This demonstrates that Trace Inversion is effective across both small- and large-scale models. We observe similar gains in abstention accuracy (see Appendix E).

| | MMLU | | | | | K-Crosswords | | | | | Hellaswag | | | | | Propaganda | | | | |
|---|---|---|---|---|---|---|---|---|---|---|---|---|---|---|---|---|---|---|---|---|
| | M | P | Q | D | G | M | P | Q | D | G | M | P | Q | D | G | M | P | Q | D | G |
| TOKENPROB | .661 | .374 | .645 | .500 | .485 | .489 | .420 | .737 | .578 | .525 | .678 | .602 | .693 | .610 | .598 | .333 | .323 | .596 | .470 | .445 |
| ASKCALI | .697 | .434 | .636 | .643 | .618 | .550 | .189 | .713 | .580 | .600 | .618 | .708 | .721 | .677 | .600 | **.608** | **.800** | .669 | **.693** | .585 |
| REFLECT | .662 | .369 | .398 | .390 | .375 | .498 | .430 | .683 | .500 | .495 | .673 | .682 | .660 | .655 | .650 | .450 | .325 | .672 | .674 | .465 |
| COOPERATE | .680 | .388 | .424 | .410 | .431 | .498 | .184 | .724 | .694 | .701 | .691 | .385 | .663 | .635 | .647 | .450 | .325 | .500 | .467 | .452 |
| COMPETE | **.837** | .431 | .681 | .701 | .695 | .569 | .542 | .724 | .608 | .611 | .812 | .721 | .729 | .717 | **.711** | .402 | .434 | .835 | .427 | .395 |
| SC | .678 | .365 | .344 | .521 | .412 | .521 | .397 | .389 | .525 | .475 | .697 | .412 | .389 | .618 | .533 | .326 | .389 | .445 | .363 | .466 |
| ATC | .710 | **.732** | .398 | .580 | .588 | .550 | .412 | .450 | .315 | .660 | .660 | .498 | .475 | .289 | .539 | .450 | .498 | .512 | .524 | .570 |
| TrInv-SE | .571 | .250 | .398 | **.899** | .590 | .500 | .412 | **.789** | .719 | .655 | .688 | .475 | .733 | **.812** | .655 | .501 | .498 | .512 | .614 | .590 |
| TrInv-LLM | .702 | .471 | .650 | .857 | .588 | .479 | **.654** | .644 | **.812** | .627 | .743 | **.783** | .769 | .649 | .688 | .457 | .516 | .421 | .400 | **.710** |
| TrInv-GROUND | .788 | .612 | **.685** | .675 | **.699** | **.602** | .497 | .787 | .525 | **.800** | **.814** | .498 | **.780** | .612 | .656 | .409 | .504 | **.929** | .524 | .688 |

| | BBQ | | | | | Misconceptions | | | | | Quail | | | | | GSM | | | | |
|---|---|---|---|---|---|---|---|---|---|---|---|---|---|---|---|---|---|---|---|---|
| | M | P | Q | D | G | M | P | Q | D | G | M | P | Q | D | G | M | P | Q | D | G |
| TOKENPROB | .725 | .710 | .447 | .568 | .578 | .714 | .239 | .393 | .609 | .511 | .726 | .322 | **.806** | .793 | .781 | .368 | .370 | .500 | .481 | .785 |
| ASKCALI | .785 | .792 | .689 | .695 | .701 | .703 | .286 | .627 | .613 | .615 | .769 | .765 | .716 | .717 | .703 | .368 | .375 | .286 | .291 | **.796** |
| REFLECT | .672 | .670 | .661 | .675 | .663 | .692 | .690 | .683 | .676 | .671 | .711 | .708 | .705 | .700 | .698 | .392 | .395 | .390 | .385 | .683 |
| COOPERATE | .669 | .671 | .526 | .530 | .535 | .696 | .702 | .603 | .607 | .611 | .774 | .793 | .762 | .763 | .758 | .420 | .427 | .398 | .403 | .407 |
| COMPETE | .759 | .755 | .254 | .293 | .268 | **.813** | .811 | .772 | .740 | .735 | **.793** | .788 | .786 | .781 | .777 | **.635** | .648 | .656 | .675 | .651 |
| SC | .704 | .365 | .344 | .521 | **.714** | .412 | .333 | .389 | .728 | .475 | .498 | .363 | .411 | .466 | .533 | .525 | .445 | .512 | .577 | .590 |
| ATC | .778 | .398 | .450 | .580 | .588 | .439 | .660 | .512 | .625 | .670 | .498 | .412 | .471 | .524 | .566 | .498 | .531 | .543 | .597 | .600 |
| TrInv-SE | .812 | .583 | .501 | **.819** | .688 | .588 | .512 | .702 | **.813** | .655 | .604 | .471 | .523 | .578 | .611 | .598 | .550 | .612 | .680 | .703 |
| TrInv-LLM | .754 | .753 | **.742** | .667 | .661 | .812 | **.827** | .574 | .748 | **.882** | .493 | .672 | .655 | .814 | .690 | .605 | .612 | .642 | **.708** | .719 |
| TrInv-GROUND | **.929** | **.812** | .657 | .677 | .700 | .784 | .529 | **.800** | .782 | **.886** | .534 | **.798** | .791 | **.848** | **.798** | .607 | **.720** | **.657** | .689 | .793 |

| TOKENPROB | ASKCALI | REFLECT | COOPERATE | COMPETE | SC | ATC | TrInv-SE | TrInv-LLM | TrInv-GROUND |
|---|---|---|---|---|---|---|---|---|---|
| 0.555 | 0.611 | 0.579 | 0.560 | 0.649 | 0.479 | 0.533 | 0.612 | 0.666 | **0.697** |

(a) Reliable accuracy (R-Acc) for each method averaged across all settings.

Table 2: Results showing how our Trace Inversion methods outperform previous abstention baselines by reliable accuracy (**R-Acc**) across five models and eight datasets. For brevity, we again use a mapping for this table where model abbreviations are as follows: M for Mistral-7B-Instruct-v0.3; P for phi-4; Q for Qwen2.5-32B; D for DeepSeek-R1-Distill-Qwen-32B; and G for gpt-oss-20b. Blue rows correspond to our Trace Inversion methods. Best results in **bold** and second best in underline. Trace Inversion methods perform the best in 28 out of 40 settings and at least top two of the ten methods in 37 out of 40 settings.

Importantly, the improvement afforded by Trace Inversion methods addresses the degradation observed when using CoT outputs in prior baselines (as indicated by cross-hatched regions in Figure 4). Unlike CoT-based predictions, Trace Inversion methods leverage inverted traces to recover the most reliable model behavior, achieving higher alignment between abstention mechanisms and model outputs. By thinking about abstention as evaluating whether the model is actually answering the *wrong* question and leveraging the information provided by reasoning traces, Trace Inversion is addressing the core problem of abstention. Overall, these results indicate that Trace Inversion provides

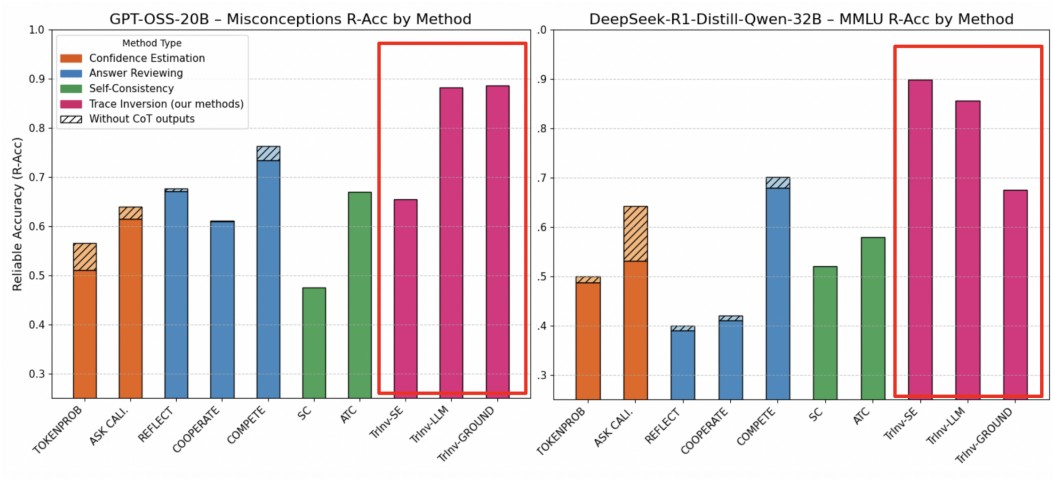

Figure 4: We highlight two of the 28 settings in which Trace Inversion methods outperform current abstention baselines. We also show the aforementioned CoT-related degradation of abstention through the cross-hatching portion.

a systematic and robust enhancement to abstention strategies, improving reliable accuracy across heterogeneous tasks and models.

## 4.2 LIMITATIONS AND FUTURE WORK

Our methods have limitations and lay the ground work for future study. First, our evaluation focuses on a variety of benchmarks but does not capture the full variety of real-world queries, such as those with false premises or temporal lags (Kirichenko et al., 2025), where abstention may behave differently. Second, we frame abstention purely in terms of knowledge gaps, without considering human-valued reasons for abstention such as safety or harm reduction with queries (Yang et al., 2024d; Zhou et al., 2024). Finally, even though we observe promising gains in abstention performance, our method depends on reconstructing queries through LLM generations, which introduces potential noise; future work could explore reconstruction methods that leverage model internals to obtain more faithful representations of the model's implicit query.

## 5 CONCLUSION

This work introduces **Trace Inversion**, a set of methods for improving model abstention by inverting reasoning traces. Across five LLMs and eight benchmark datasets, Trace Inversion outperformed state-of-the-art abstention baselines, demonstrating its robustness to variety of tasks and domains. We also propose a new framework for understanding hallucinations: rather than treating them as models answering questions incorrectly, we frame them as models answering the *wrong* question. This contribution suggests several avenues for future research, including the development of methods to probe misaligned internal reasoning, the design of training objectives paradigms that minimize exploration of spurious reasoning paths, and the creation of evaluation benchmarks that capture subtle errors in reasoning alignment. Finally, our findings reveal that Chain-of-Thought outputs can sometimes degrade abstention baselines; Trace Inversion counteracts this issue by repurposing reasoning traces, ultimately turning them into a source of strength for abstention.

## REPRODUCIBILITY STATEMENT

We have taken steps in this work to ensure the reproducibility of our results. All models and datasets used in our experiments are available and we release the complete source code. In the main paper and appendices material, we provide complete details of all experimental setups, including model architectures and hyperparameters. We believe that the measures we have taken to ensure repro-

ducibility will facilitate straightforward replication and verification of our findings, as well as allow the community to build upon our results in the future.

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
