## APPENDIX A    LLM PROMPTS

We provide the exact prompts we employed in the study in the order they first appear.

> **ASKCALI**    Provide your best guess for the following question. Give ONLY the guess, no other words or explanation.
>
> For example:
> Guess: {most likely guess, as short as possible; not a complete sentence, just the guess!}
> The question is: {question}
>
> [*LLM-generated answer*]
> Provide the probability that your guess is correct. Give ONLY the probability, no other words or explanation.
>
> For example:
>
> Probability: {the probability between 0.0 and 1.0 that your guess is correct, without any extra commentary whatsoever; just the probability!}
> Probability: [*LLM-generated probability*]

> **REFLECT**    {question}
>
> [*LLM-generated answer*]
> The above answer is:
> A. True
> B. False
> The answer is [LLM-generated A/B].

> **COOPERATE**    Question: {question}
>
> Answer: [*LLM-generated proposed answer*]
>
> for domain in ["factual information", "commonsense knowledge", "mathematical knowledge"]:
>     Generate some knowledge about the question, focusing on {domain}
>     Knowledge: {generated domain knowledge}
>     Question: {question}
>     Answer: {generated proposed answer}
>
> Please review the proposed answer and provide feedback on its correctness. Feedback: [generated feedback]
>
> Question: {question}
> Proposed Answer: {generated proposed answer}
>
> Feedback 1: {generated feedback from expert 1}
> ...
> Feedback k: {generated feedback from expert k}
> Based on the feedback, the proposed answer is:
> A. True
> B. False
> The answer is [*LLM-generated A/B*].

**COMPETE**    Question: {question}

Answer: [*LLM-generated proposed answer*]

if multiple-choice:
    {alternative answer} = randomly select another unchosen answer
else:
    Question: {question}
    Answer: {generated proposed answer}
    Please propose an alternative answer: [*alternative answer*]

Question: {question}
Generate a knowledge paragraph about {alternative answer}: [*generated alternative passage*]

Answer the question with the following knowledge:  feel free to ignore irrelevant or wrong information.

Knowledge: {generated alternative passage}
Question: {question}
Answer: [*new generated answer*]

if {new generated answer} == {generated proposed answer}:
    abstain = False
else:
    abstain = True

We now provide the prompts used in our trace inversion procedure. First, we detail the prompt used for reconstructing the model query below.

**Query Reconstruction Prompt**    You are a puzzle solver.  Given the following reasoning trace, reconstruct the initial question by interpreting the steps in the reasoning trace. Do not answer the question.
Reasoning Trace:
{reasoning trace}

Reconstructed query:

Now, we provide the prompt used in the `TrInv-LLM` method.

> **TrInv-LLM**  Do the following two prompts convey the same framing?
>
> Prompt 1: {q1}
> Prompt 2: {q2}
>
> Select YES or NO:
> Final answer:
>
> Do the following two prompts convey the same intent?
>
> Prompt 1: {q1}
> Prompt 2: {q2}
>
> Select YES or NO:
> Final answer:
>
> Do the following two prompts convey the same details as context?
>
> Prompt 1: {q1}
> Prompt 2: {q2}
>
> Select YES or NO:
> Final answer:

## APPENDIX B    ADDITIONAL BASELINES

Another means of measuring model confidence is to evaluate the consistency of reasoning traces. We use two approaches of this flavor.

**Self-consistency (SC)**    Self-consistency is a method that samples multiple reasoning paths and aggregates final answers through majority voting (Wang et al., 2022). Based on a previous abstention study (Feng et al., 2024), we calculate a plurality index as the confidence score $p_i$. Given a question $q_i$ along with the $n$ generated reasoning paths and final answers $a_i = \{a_{ij}\}_{j=1,...,n}$, the plurality index is defined as:

$$p_i = \text{plu}(q_i, a_i, n) = \max_{a_{ij}} \sum_{t=1}^{n} \mathbf{1}(a_{ij} = a_{it})$$

**Average trace confidence (ATC)**    Inspired by self-consistency, several approaches like DeepConf (Fu et al., 2025b) have explored average trace confidence (also termed self-certainty) (Kang et al., 2025) as a trace-level quality measure. Here, we use average trace confidence over $n$ reasoning paths as the confidence score $p_i$. First, we calculate the token confidence as the sum of the negative average log-probability of the top-$k$ tokens considered at position $\ell$ where $P_\ell$ is the language model's predicted token distribution at index $\ell$.

$$\text{tok} = -\frac{1}{k} \sum_{t=1}^{k} \log P_\ell(t)$$

Then, for each reasoning trace $r_{ij}$, trace confidence is the average token confidence over all $N$ generated tokens

$$C_{ij} = \frac{1}{N} \sum_{t=1}^{N} \text{tok}_t$$

Finally, to calculate the average trace confidence over $n$ reasoning paths for question $q_i$, we have the confidence score

$$p_i = \frac{1}{n} \sum_{j=1}^{n} C_{ij}$$

## APPENDIX C   EXPERIMENTAL SETTINGS

### C.1   MODEL PARAMETERS

**Model Initialization.**   We support multiple large language models (LLMs) through a unified initialization function. The implementation maps human-readable names (e.g., `mistral`, `llama2_70b`, `qwen_32b`) to their respective HuggingFace or vLLM model checkpoints. Models are loaded with `bfloat16` precision and GPU memory utilization capped at 80% for efficiency. Chat-oriented models (e.g., DeepSeek, Qwen, Mistral) are automatically wrapped with their tokenizer's chat template. Our code also enables easy integration of new models.

**Sampling Parameters.**   Responses are generated with configurable temperature ($T = 0.1$ by default), a maximum of 1024 new tokens, and optional token-level probabilities. The code supports exponential backoff retries (up to 10 attempts) to ensure robustness against API or inference errors.

**Answer Parsing.**   Since models may return heterogeneous outputs, we implement rule-based answer parsing with multiple heuristics (e.g., "Answer: A", "The correct answer is B", or isolated multiple-choice options). Unparseable responses are labeled with a sentinel "Z" to indicate incorrectness.

### C.2   DATASETS

We elaborate on the eight datasets used. Each sample question contains multiple choice answers and corresponding metadata, such as bias type for BBQ or reading comprehension task for Quail. All datasets used can be found in our Github repo.

1. MMLU is a multiple-choice dataset for general knowledge QA including elementary mathematics, US history, computer science, law, and more (Hendrycks et al., 2021).

2. Knowledge Crosswords (K-Crosswords) is a geometric knowledge reasoning benchmark consisting of incomplete knowledge networks bounded by structured factual constraints (Ding et al., 2024).

3. Hellaswag is dataset that tests commonsense natural language inference (Zellers et al., 2019).

4. Propaganda dataset tasks LLMs with identifying the 23 persuasion tactics in a long news article based on their internal knowledge (Piskorski et al., 2023).

5. Bias Benchmark for Question Answering (BBQ) is a dataset of question sets constructed by the authors that highlight attested social biases against people belonging to protected classes along nine social dimensions relevant for U.S. English-speaking contexts (Parrish et al., 2022).

6. 'Misconceptions' task also from BIG-Bench measures whether a model can discern popular misconceptions from the truth (Srivastava et al., 2023).

7. Quail is a reading comprehension dataset containing answerable and unanswerable passage-based questions (Rogers et al., 2020).

8. GSM-MC (Zhang et al., 2024) is a multiple-choice dataset constructed by collecting answers and incorrect predictions on GSM8K (Cobbe et al., 2021) from 60 open-source models.

We also provide summary statistics illustrating the size of dataset and question length distribution (see Table 3).

| Dataset | Total | Answerable (%) | Avg Q Len | Med Q Len | Avg Choices |
|---------|-------|----------------|-----------|-----------|-------------|
| MMLU | 2,000 | 100.0 | 204 | 97 | 4.0 |
| K-Crosswords | 2,101 | 100.0 | 403 | 399 | 4.0 |
| Hellaswag | 2,000 | 100.0 | 223 | 238 | 4.0 |
| Propaganda | 431 | 100.0 | 4273 | 4384 | 4.0 |
| BBQ | 900 | 50.0 | 248 | 228 | 2.0 |
| Misconceptions | 219 | 100.0 | 83 | 70 | 2.0 |
| Quail | 3,000 | 88.9 | 1966 | 1936 | 4.0 |
| GSM | 3,000 | 100.0 | 236 | 216 | 4.0 |

Table 3: Summary statistics on eight datasets and specific samples we used for our results. Average question length and median question length are word counts of only the question sans prompt in each dataset.

## APPENDIX D  VERBOSITY OF TRACE GENERATION

To better understand the potential verbosity or redundancy present in generated traces versus standard outputs, we measured differences in word length, number of sentences, and repetition ratio. Repetition ratio (Rep) is measured as the 1 - (number of unique words / by the total number of words) (see Table 4). We also include the number of reasoning steps in the last column of the table to show model and dataset level differences.

| Model | Dataset | Std Words | CoT Words | Word △ | Std Sents | CoT Sents | Sent △ | Std Rep | CoT Rep | Steps |
|-------|---------|-----------|-----------|--------|-----------|-----------|--------|---------|---------|-------|
| phi-4 | MMLU | 744.07 | 727.81 | -16.26 | 3.33 | 4.48 | +1.14 | 0.52 | 0.65 | 9.39 |
| | K-Crosswords | 620.73 | 604.12 | -16.61 | 4.19 | 11.02 | +6.83 | 0.62 | 0.65 | 8.50 |
| | Hellaswag | 831.58 | 768.12 | -63.46 | 3.36 | 4.13 | +0.76 | 0.57 | 0.61 | 10.66 |
| | Propaganda | 511.85 | 696.55 | +184.70 | 2.39 | 3.92 | +1.53 | 0.41 | 0.53 | 10.40 |
| | Misconceptions | 426.88 | 772.88 | +346.00 | 1.56 | 4.08 | +2.51 | 0.34 | 0.59 | 16.50 |
| | Quail | 204.20 | 742.10 | +537.90 | 1.49 | 9.22 | +7.73 | 0.17 | 0.58 | 12.01 |
| | GSM | 774.43 | 576.13 | -198.30 | 3.82 | 6.91 | +3.09 | 0.57 | 0.67 | 10.27 |
| | BBQ | 187.28 | 704.89 | +517.61 | 0.86 | 4.06 | +3.21 | 0.12 | 0.63 | 10.39 |
| Qwen2.5-32B | MMLU | 61.95 | 153.68 | +91.73 | 3.52 | 10.90 | +7.38 | 0.22 | 0.46 | 4.59 |
| | K-Crosswords | 76.49 | 242.16 | +165.67 | 5.72 | 14.27 | +8.55 | 0.23 | 0.66 | 5.07 |
| | Hellaswag | 30.52 | 134.50 | +103.98 | 1.88 | 9.63 | +7.75 | 0.10 | 0.42 | 4.89 |
| | Propaganda | 59.15 | 107.45 | +48.30 | 2.85 | 6.05 | +3.20 | 0.19 | 0.34 | 1.75 |
| | Misconceptions | 34.25 | 92.63 | +58.38 | 2.75 | 7.88 | +5.13 | 0.10 | 0.34 | 3.38 |
| | Quail | 34.10 | 128.96 | +94.86 | 2.21 | 9.28 | +7.07 | 0.13 | 0.39 | 4.27 |
| | GSM | 123.00 | 143.39 | +20.39 | 10.03 | 11.22 | +1.19 | 0.45 | 0.56 | 6.58 |
| | BBQ | 36.67 | 153.89 | +117.22 | 2.33 | 10.44 | +8.11 | 0.17 | 0.44 | 4.78 |
| gpt-oss-20b | MMLU | 351.14 | 703.88 | +352.74 | 3.45 | 8.98 | +5.53 | 0.87 | 0.51 | 3.62 |
| | K-Crosswords | 631.72 | 653.96 | +22.24 | 7.89 | 7.95 | +0.06 | 0.81 | 0.73 | 0.42 |
| | Hellaswag | 463.06 | 756.76 | +293.70 | 4.10 | 8.74 | +4.64 | 0.89 | 0.57 | 3.32 |
| | Propaganda | 613.15 | 663.05 | +49.90 | 5.50 | 8.66 | +3.16 | 0.82 | 0.57 | 1.85 |
| | Misconceptions | 218.75 | 744.00 | +525.25 | 2.11 | 8.55 | +6.44 | 0.87 | 0.48 | 4.63 |
| | Quail | 414.21 | 716.83 | +302.62 | 4.32 | 10.18 | +5.86 | 0.87 | 0.56 | 3.21 |
| | GSM | 245.67 | 628.19 | +382.52 | 3.22 | 10.85 | +7.62 | 0.82 | 0.50 | 4.06 |
| | BBQ | 533.06 | 748.22 | +215.17 | 5.12 | 10.69 | +5.58 | 0.90 | 0.68 | 2.56 |

Table 4: Summary statistics of CoT verbosity compared to standard outputs.

We observe that across all models, the CoT repetition ratio is higher than the standard output repetition ratio. As expected, there are both more words and sentences in CoT outputs.

## APPENDIX E  ADDITIONAL RESULTS

Below, we provide the abstention accuracy results across all methods, datasets, and models with and without CoT outputs. Table 5 shows that cross nearly all baselines, the CoT variants (highlighted in red) tend to exhibit a consistent degradation in abstention accuracy (**A-Acc**). For example, `Tr-TOKENPROB` shows a marked drop on datasets like K-Crosswords and Hellaswag. Similarly, `Tr-REFLECT` and `Tr-COOPERATE` also experience substantial drops in certain datasets. For instance, `Tr-REFLECT` sees a decrease on Quail for Qwen2.5-32B from 0.651 to 0.582 (-0.069) and on GSM from 0.605 to 0.587 (-0.018), showing that even baselines designed to simulate internal deliberation are poor at evaluating additional reasoning steps.

The results in Table 6 highlight the substantial improvements offered by the Trace Inversion methods over traditional abstention baselines in terms of abstention accuracy. In the top portion of the

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

Table 5: Results showing degradation of abstention baselines with CoT outputs. This table shows reduced reliable accuracy (**A-Acc**) across five models and eight datasets for each of the five abstention baselines. For brevity, we use a mapping for this table where model abbreviations are as follows: M for Mistral-7B-Instruct-v0.3; P for phi-4; Q for Qwen2.5-32B; D for DeepSeek-R1-Distill-Qwen-32B; and G for gpt-oss-20b. Red rows indicate use of CoT outputs. **Bold** values indicate the higher performance between the baseline and CoT variant.

table, we see that conventional methods such as TOKENPROB, ASKCALI, REFLECT, COOPERATE, and COMPETE generally achieve moderate reliable accuracy (**A-Acc**), with considerable variability depending on dataset. Similarly, ATC and SC provide improvements on some datasets but fail to consistently achieve high performance, highlighting the limitations of conventional abstention approaches in capturing when a model's prediction may be unreliable across abstention scenarios.

In contrast, the Trace Inversion methods consistently outperform these baselines across nearly every dataset. For example, TrInv-GROUND reaches the highest abstention accuracy in datasets such as BBQ (.930 for Mistral-7B), MMLU (.786 for Deepseek-Distill-Qwen-32B), and GSM (.795 for gpt-oss-20b), demonstrating its robust ability to detect uncertain predictions. Even in cases where TrInv-GROUND is not the top performer, the other Trace Inversion methods (TrInv-SE or TrInv-LLM) often rank in the top two, indicating that the Trace Inversion approach reliably identifies uncertainty across models and tasks.

Notably, Trace Inversion methods appear to scale well with model capability: larger or more sophisticated models, such as Qwen2.5-32B and gpt-oss-20b, show marked gains in A-Acc when using Trace Inversion, whereas traditional methods fail to capitalize on these improvements. Overall, the table illustrates that Trace Inversion provides a systematic and robust mechanism for abstention, outperforming existing baselines in 28 out of 40 settings and ranking in the top two in 37 out of 40 settings, highlighting its generalizability and effectiveness across a wide range of models and tasks.

| | MMLU | | | | | K-Crosswords | | | | | Hellaswag | | | | | Propaganda | | | | |
|---|---|---|---|---|---|---|---|---|---|---|---|---|---|---|---|---|---|---|---|---|
| | M | P | Q | D | G | M | P | Q | D | G | M | P | Q | D | G | M | P | Q | D | G |
| TOKENPROB | .657 | .477 | **.743** | .603 | .528 | .487 | .312 | **.721** | .552 | .498 | .678 | .624 | **.663** | .642 | .618 | .353 | .402 | .565 | .497 | .492 |
| ASKCALI | .675 | .471 | .604 | .735 | .772 | .579 | .191 | .398 | .519 | .647 | .660 | .477 | .593 | .624 | .631 | **.685** | .598 | .503 | .547 | .522 |
| REFLECT | .655 | .379 | .457 | .602 | .523 | .501 | .308 | .446 | .553 | .497 | .667 | .621 | .576 | .602 | .583 | .352 | .402 | .563 | .573 | .498 |
| COOPERATE | .656 | .422 | .537 | .603 | .552 | .504 | .271 | .588 | .631 | .598 | .591 | .392 | .586 | .587 | .603 | .452 | .231 | .527 | .573 | .476 |
| COMPETE | .675 | .571 | .705 | .703 | .652 | .572 | .304 | .417 | .553 | .502 | .536 | .503 | .486 | .552 | .523 | .587 | .552 | .451 | .498 | .482 |
| SC | .677 | .365 | .352 | .543 | .412 | .527 | .411 | .481 | .498 | .503 | .683 | .439 | .492 | .506 | .515 | .335 | .380 | .447 | .362 | .410 |
| ATC | **.710** | **.841** | .409 | .490 | .588 | .511 | .437 | .468 | .315 | .640 | .634 | .498 | .475 | .289 | .430 | .450 | .498 | .512 | .524 | .570 |
| TrInv-SE | .310 | .663 | .418 | .460 | .432 | .585 | .412 | .620 | .471 | .533 | .402 | **.698** | .573 | .482 | .612 | .467 | .498 | **.712** | **.614** | .590 |
| TrInv-LLM | .540 | .620 | .625 | .739 | **.858** | **.685** | **.751** | .318 | .537 | .562 | **.713** | .690 | .390 | .655 | .571 | .662 | **.775** | .475 | .550 | **.603** |
| TrInv-GROUND | .479 | .508 | .632 | **.789** | .801 | .506 | .478 | .572 | **.711** | **.700** | .712 | .504 | .540 | **.695** | **.700** | .498 | .620 | .500 | .503 | .500 |

| | BBQ | | | | | Misconceptions | | | | | Quail | | | | | GSM | | | | |
|---|---|---|---|---|---|---|---|---|---|---|---|---|---|---|---|---|---|---|---|---|
| | M | P | Q | D | G | M | P | Q | D | G | M | P | Q | D | G | M | P | Q | D | G |
| TOKENPROB | .657 | .477 | .743 | .603 | .528 | .487 | .312 | .721 | .552 | .498 | **.678** | .624 | .663 | .642 | .618 | .353 | .402 | .565 | .497 | .492 |
| ASKCALI | .675 | .471 | .604 | .735 | **.772** | .579 | .191 | .398 | .519 | .647 | .660 | .477 | .593 | .624 | .631 | **.685** | .598 | .503 | .547 | .522 |
| REFLECT | .655 | .379 | .457 | .602 | .523 | .501 | .308 | .446 | .553 | .497 | .667 | .621 | .576 | .602 | .583 | .352 | .402 | .563 | .523 | .498 |
| COOPERATE | .656 | .422 | .537 | .603 | .552 | .504 | .271 | .588 | .631 | .598 | .591 | .392 | .586 | .587 | .603 | .452 | .231 | .527 | .573 | .476 |
| COMPETE | .675 | .571 | .705 | .703 | .652 | .572 | .304 | .417 | .553 | .502 | .536 | .503 | .486 | .552 | .523 | .587 | .552 | .451 | .698 | .682 |
| SC | .706 | .380 | .514 | .523 | .728 | .427 | .361 | .390 | .729 | .481 | .499 | .366 | .412 | .469 | .536 | .528 | .447 | .513 | .579 | .593 |
| ATC | .718 | .402 | .452 | .583 | .591 | .442 | .662 | .514 | .628 | .673 | .500 | .415 | .473 | .527 | .569 | .500 | .533 | .545 | .599 | .603 |
| TrInv-SE | .813 | .585 | .502 | **.900** | .689 | .590 | .515 | .705 | **.815** | .657 | .606 | .473 | .524 | .580 | .613 | .600 | .552 | .615 | .682 | .705 |
| TrInv-LLM | .755 | .755 | .744 | .669 | .663 | **.814** | **.829** | .575 | .750 | .885 | .495 | .674 | .657 | .816 | .692 | .607 | .614 | .644 | **.710** | .721 |
| TrInv-GROUND | **.930** | **.814** | .658 | .679 | .701 | .786 | .531 | **.802** | .784 | **.888** | .536 | **.800** | **.793** | **.850** | **.800** | .609 | **.722** | **.659** | .690 | **.795** |

Table 6: Results showing how our Trace Inversion methods outperform previous abstention baselines by abstention accuracy (**A-Acc**) across five models and eight datasets. For brevity, we again use a mapping for this table where model abbreviations are as follows: M for Mistral-7B-Instruct-v0.3; P for phi-4; Q for Qwen2.5-32B; D for DeepSeek-R1-Distill-Qwen-32B; and G for gpt-oss-20b. Best results in **bold** and second best in underline.