# OpenReview forum: "Chain-of-Thought Degrades Abstention in Large Language Models, Unless Inverted"
_ICLR.cc/2026/Conference — ICLR 2026 Conference Withdrawn Submission_

### Official Review · Reviewer_fpXh · 2025-10-15

**Soundness:** 3
**Presentation:** 3
**Contribution:** 2
**Rating:** 4
**Confidence:** 4

**Summary:**

This paper explores the performance of abstention methods when including Chain-of-Thought (CoT) traces. The authors evaluate several baseline abstention methods on 8 datasets and using 5 open-weights models, and find that including CoT traces generally worsens abstention. They then introduce Trace Inversion, a new abstention method that regenerates a prompt given the current reasoning trace, and then uses distance between the original prompt and the regenerated prompt as an abstention metric, where high distances should be abstains. The authors test three different ways of measuring prompt distance, and find promising results for their approach overall.

**Strengths:**

* The paper's empirical observation that passing CoT reasoning traces into abstention methods worsens their performance is important.
* The idea behind "Trace Inversion", i.e., to regenerate the query from the sampled reasoning trace and compute its distance from the original query as an abstention signal, is novel, and the results seem promising overall.
* The authors select a wide variety of baseline methods, and evaluate on 8 datasets and with 5 open-weights models.

**Weaknesses:**

* The authors present very large results tables and only limited aggregated statistics, which I fear may obfuscate the scale of the problem. Table 1, for example, has no mean accuracy over models or methods, while Table 2 has the mean per method. Particularly given the variability of the results (i.e., there's no clear winner), finding a more appropriate way to present the high-level results---either as a table of aggregated results, or as a figure---would make this paper much stronger.
* The authors test three different implementations of Trace Inversion using different distance measures. However, there is no clear winner as to which distance measure works best. In Figure 4, for example, the best-performing method for GPT-OSS on Misconceptions is the worst for DeepSeek-Distill-Qwen on MMLU. How should someone who wants to apply Trace Inversion to a model decide which to use?
* "Reliable Accuracy" seems improperly specified, and would reward over-abstention (a trivial example would be abstaining for all but one answerable questions). A more appropriate choice, given the threshold-based formulation in section 3.1, would be something like an Accuracy-Rejection Curve.
* No large-scale, API models are evaluated. While this is perhaps understandable given cost constraints, several of the abstention methods provided could be used on closed models. This makes it hard to understand whether the results hold on large-scale frontier models, and should at least be discussed as a limitation.

**Questions:**

1. Kirichenko et al. [1] have previously reported that including CoT traces can inflate the performance of an LLM-as-judge abstention detector. How do you reconcile this result with your finding that including the trace worsens abstention, in particular for the (relatively similar) REFLECT method?
2. How is the correctness of final answers verified for the non-multiple-choice benchmarks? Exact match or some other approach?
3. A threshold parameter is introduced on line 179. How is this applied for methods that don't emit a probability, such as REFLECT?
4. On the datasets which only contain answerable questions, the authors state that "the model is expected to abstain when it does not have the knowledge to answer" (line 259). How is this determined, and when should the models actually abstain for these 6 datasets?


References

[1] https://arxiv.org/abs/2506.09038

---

> ### Author Response · Authors · 2025-11-22
> **Response to Reviewer fpXh**
>
> We thank the reviewer for the positive assessment of our empirical findings, the novelty of Trace Inversion, and the breadth of our evaluations across models, datasets, and baseline methods. We address the concerns below and will incorporate the suggested improvements into the final version.
>
> The reviewer notes that our results tables are large and that limited aggregation may obscure high-level patterns. We agree that providing clearer summaries will strengthen the paper. In the revision, we will include an aggregate table that consolidates mean abstention accuracy and reliable accuracy across models and methods for each dataset (see Table 1). This will complement the detailed tables by making the overall trends more easily identifiable, especially given that the best method varies across settings. We will also consider adding a visualization summarizing these aggregates, which we expect to significantly improve readability.
> Regarding the choice among the three Trace Inversion variants, we appreciate the request for clearer guidance. Each variant captures a different type of mismatch between the original query and the reconstructed query, and their strengths vary accordingly:
>
> - TrInv-GROUND is most effective at detecting semantic shifts of intent, especially in social, bias, and safety-related domains where subtle changes in presuppositions matter.
> - TrInv-SE (sentence embeddings) excels when the reconstructed trace introduces additional assumptions or details prevalent in mathematical and analytical reasoning tasks.
> - TrInv-LLM, which evaluates groundedness directly using a language model, performs well across a blend of domains and shows promise as a robust general-purpose option.
>
> To make this clearer to practitioners, our revision will include an aggregate table (see Table 1) summarizing the best-performing Trace Inversion variant for each model–dataset pair, along with general guidelines illustrating the benefit of each trace inversion variant in their optimal usage scenario.
>
> The reviewer raises a thoughtful point about “Reliable Accuracy” potentially rewarding over-abstention. To avoid this issue, we emphasize that Reliable Accuracy is not used for threshold tuning. Thresholds for all abstention methods are selected using Abstention Accuracy (ACC) [1]. This metric penalizes both incorrect answers and inappropriate abstentions, preventing degenerate solutions such as abstaining on nearly all queries. We provide full abstention accuracy curves in Appendix E, and in the revision we will explicitly clarify that ACC governs threshold selection.
>
> The reviewer also notes the absence of evaluations on large-scale API frontier models. We agree this is an important limitation. Cost constraints make extensive abstention sweeps difficult for proprietary APIs, particularly because many abstention methods require revealing chain-of-thought (which is not always accessible). Instead, we evaluate on GPT-OSS and other open-weights models that provide controllable reasoning traces and are feasible for systematic study. We will include a discussion acknowledging this limitation and clarifying that Trace Inversion is model-agnostic and naturally applicable in black-box settings, since the abstention signal derives solely from the sampled reasoning trace and the reconstructed query.
>
> We thank the reviewer again for the constructive suggestions.
>
> Table 1: Aggregated Table
> | Method        | M     | P     | Q     | D     | G     |
> |---------------|-------|-------|-------|-------|-------|
> | TOKENPROB     | 0.560 | 0.403 | 0.616 | 0.580 | 0.584 |
> | ASKCALI       | 0.657 | 0.558 | 0.627 | 0.645 | 0.663 |
> | REFLECT       | 0.573 | 0.508 | 0.596 | 0.570 | 0.561 |
> | COOPERATE     | 0.593 | 0.481 | 0.563 | 0.538 | 0.548 |
> | COMPETE       | 0.605 | 0.618 | 0.668 | 0.629 | 0.607 |
> | SC            | 0.529 | 0.388 | 0.396 | 0.539 | 0.523 |
> | ATC           | 0.604 | 0.503 | 0.486 | 0.501 | 0.589 |
> | **TrInv-SE**      | 0.562 | 0.450 | 0.564 | **0.711** | 0.619 |
> | **TrInv-LLM**     | 0.629 | 0.627 | 0.566 | 0.680 | 0.673 |
> | **TrInv-GROUND**  |**0.675** | **0.647** | **0.673** | 0.625 | **0.703** |
>
> [1] Wen, Bingbing, et al. "Know your limits: A survey of abstention in large language models." *Transactions of the Association for Computational Linguistics* 13 (2025): 529-556.

---

> ### Author Response · Authors · 2025-11-26
>
> Thank you again for your thoughtful initial feedback. We wanted to kindly check in to see if any additional clarification would be helpful. We are happy to provide further details or address any outstanding concerns.

---

### Official Review · Reviewer_D1CA · 2025-10-30

**Soundness:** 2
**Presentation:** 3
**Contribution:** 2
**Rating:** 2
**Confidence:** 4

**Summary:**

The authors propose a new method called Trace Inversion to determine whether a large language model (LLM) answered the intended question in order to abstain from providing a response. The authors also benchmark several baseline abstention methods including confidence-based and answer reviewing approaches, showing including reasoning traces in the model responses degrades abstention. The authors benchmark eight datasets including commonly used benchmarks such as MMLU across several open models including DeepSeek R1, Qwen, and GPT OSS.

**Strengths:**

The authors explore the important problem of abstention, particularly for models that produce chain-of-thought reasoning traces. The authors include a reasonble set of baseline methods includign confidence based as well as answer-reviewing methods. The authors confirm prior findings that reasoning degrades abstention.

The proposed method, Trace Inversion, is clear and well presented. The method covers the issue of models responding to the wrong query using self-review method and response similarity. The authors show Trace Inversion, depending on the choice of similarity between the original and reconstructed query, can outperform existing methods

I appreciate the authors’ inclusion of the code, integration with VLLM, and straightforward README.

**Weaknesses:**

# Scope of claim is far out of what is reasonably supported by experiments

- The author's frame abstention as "“Hallucinations are the result of models answering a different question that the intended one.” While this is part of abstention, models can also hallucinate context or facts while answering the intended query. The authors inappropriately recast all of abstention under this narrow umbrella.

# Experimental Setup

- Table 1 missing baseline abstention without abstention methods. This is key to verify that these method do in fact boost abstention and that they provide reasonable baselines against which to compare Trace Inversion.
- Choice of datasets is suspect? Table 3 shows only 2 out of the 8 datasets used in evaluation have unanswerable questions, which is not the best setting for evaluating abstention (the ability to recognize unanswerable questions).

# Trace Inversion's gains are not systematic or robust

- Claim on line 453 “Trace Inversion provides a systematic and robust enhancement to abstention strategies” is not reflected in experimental results. The best Trace Inversion method varies by model and by benchmark—and trace inversion is in fact not always the best depending on the similarity metric compared to baselines as shown in table 1. In some cases inversion methods are worse than uncertainty or answer reviewing baselines. For example, Figure 4: Trace Inversion Ground is tied for best for GPT-OSS but is worse than baselines for DeepSeek R1

# Computational cost

- The method is quite costly as we have to run inference then process all the chain fo thought again to regenerate the question adn compare the similar fo the generated and original query.  This isn't discussed or measured in the paper.

**Questions:**

see above

---

> ### Author Response · Authors · 2025-11-22
> **Response to Reviewer D1CA**
>
> We thank the reviewer for the thoughtful feedback and for highlighting the clarity of Trace Inversion and the quality of our released code. We address the concerns below and will incorporate the suggested revisions.
>
> The reviewer notes that our framing is too narrow: hallucinations arise when the model answers a different question than the intended one. We appreciate this point and will revise the wording to make clear that Trace Inversion targets a specific and common failure mode, rather than claiming that all hallucinations stem from misinterpreted queries. Models can certainly hallucinate facts or context even when the query is understood; Trace Inversion is explicitly designed to mitigate errors arising from the model’s internal reasoning introducing implicit assumptions not present in the original question, as illustrated in Fig. 3.
>
> Regarding the experimental setup, we agree it is useful to include previous abstention rates prior to method application. We would like to clarify that all methods are applied explicitly when there is no abstention detected from the model response. All abstention methods in our evaluation operate post-sampling. We will make this explicit and ensure clarity.
>
> The reviewer questions whether our datasets sufficiently evaluate abstention, since only two contain explicitly unanswerable questions. Our goal is broader than detecting truly unanswerable queries: many benchmarks we use, such as Misconceptions and PropagandaQA, contain questions that require domain-specific or specialized knowledge, where models frequently produce confident but unsupported answers [1]. These settings provide meaningful opportunities to measure abstention on unknown or knowledge-sparse queries, not only formally unanswerable ones. Additionally, we would like to emphasize that our unanswerable datasets BBQ and Quail were particularly curated for this setup. We will revise the text to clarify this distinction and align expectations. For example, BBQ's templated structure in its prompts is intentionally aimed at eliciting stereotype-related bias through exploiting under-specification. We will revise the text to clarify this distinction and align expectations.
>
> We also appreciate the reviewer’s point to bring up our claim on line 453: Trace Inversion provides a “systematic and robust” improvement. It is correct that the best-performing Trace Inversion variants differ across models and datasets, and in some cases strong baselines outperform certain inversion variants. However, it is clear that in most scenarios our method has advantages beyond baselines. Each variant of our Trace Inversion methods provide unique advantages for different scenarios; TrInv-GROUND performs well in detecting semantic shifts of intent, particularly for social and bias settings. TrInv-SE performs well in detecting additional assumptions in mathematical and reasoning domains. To make this clearer to readers, we will add an aggregate table summarizing the best-performing method for each model–dataset pair.
>
> Finally, we acknowledge the computational cost of Trace Inversion as only adding 2 additional inference calls per query. As inference-time infrastructure continues to improve, we believe this cost is justified by the safety and reliability gains.
>
> We thank the reviewer again for the constructive feedback and for recognizing the strengths of the work. We believe the revisions described above will resolve the concerns and significantly strengthen the clarity and positioning of the paper.
>
> [1] Feng, Shangbin, et al. "Don't hallucinate, abstain: Identifying LLM knowledge gaps via multi-LLM collaboration." *arXiv preprint arXiv:2402.00367* (2023).

---

> ### Author Response · Authors · 2025-11-26
>
> Thank you again for your thoughtful initial feedback. We wanted to kindly check in to see if any additional clarification would be helpful. We are happy to provide further details or address any outstanding concerns.

---

### Official Review · Reviewer_51RG · 2025-11-01

**Soundness:** 2
**Presentation:** 3
**Contribution:** 3
**Rating:** 2
**Confidence:** 5

**Summary:**

This paper investigates how Chain-of-Thought (CoT) reasoning affects LLM abstention and proposes "Trace Inversion", a method that reconstructs the query from its reasoning trace, then flags abstention when this differs from the original query. They evaluate across 8 datasets and 5 models, showing improvements in 28/40 settings.

**Strengths:**

1. Strong Empirical Results: Consistent improvements across diverse models and domains, with significant gain in reliable accuracy across 40 evaluation settings.
2. Comprehensive Evaluation: Thorough testing on multiple model families (7B to 32B parameters), diverse domains (math, reading comprehension, bias detection), and various abstention baselines.

**Weaknesses:**

1. The observation that CoT harms abstention is intuitive and not novel. Prior work cited and not cited already demonstrated that reasoning models degrade in abstention ability significantly.
2. The paper's two parts (CoT degrades abstention and Trace Inversion improves abstention) lack logical connection. The authors appear to have concatenated two separate observations without establishing why observing CoT degradation motivates the specific design of Trace Inversion. The method would work regardless of whether CoT helps or hurts.
3. The paper shows that Trace Inversion improves performance but provides no insight into why. More alabtion studies or failure / success case analysis would be helpful.

**Questions:**

See weakness.

---

> ### Author Response · Authors · 2025-11-22
> **Response to Reviewer 51RG**
>
> We thank the reviewer for the positive assessment of our empirical results and evaluation breadth, and we address the concerns below.
>
> The reviewer notes that the observation that CoT can harm abstention is intuitive and has appeared in prior work. Our goal, however, is not to claim novelty in the general phenomenon but to systematically study how specific abstention methods degrade under Chain-of-Thought conditions, especially for the models and prompting paradigms used today. Prior abstention and selective prediction methods were typically developed for settings where the model produces only a final answer. As we show, many of these methods interact poorly with CoT-style prompting, which is now standard practice in modern instruction-tuned models. Our contribution is therefore the first comprehensive, controlled comparison of abstention baselines in CoT vs. non-CoT settings across model families, which we believe is an essential step for understanding abstention behavior in current models that transcends prior assumptions.
>
> The reviewer also raises an important point about the connection between our two contributions: (1) analyzing CoT-driven abstention degradation and (2) introducing Trace Inversion. We will clarify this connection in the revision. As described at the beginning of Section 4 (L326–328), our motivation is that once we observe that CoT impairs abstention, we naturally look to the traces themselves rather than the final outputs to recover reliable alignment signals. That is, if CoT reasoning is already available and is the source of performance degradation, then using that reasoning as evidence for reconstructing the intended question is a targeted design choice, not an arbitrary second contribution. Trace Inversion is specifically motivated by leveraging the presence of CoT traces to restore cautious behavior without discarding the accuracy benefits that CoT provides.
>
> The reviewer further notes that although Trace Inversion improves performance, the paper provides limited insight into why it works. We agree this is a valuable direction and have already included an illustrative mechanism in Figure 3 (L344–357). This example walks through a real instance where the model’s internal reasoning contains additional implicit assumptions that are absent from the original query; Trace Inversion reconstructs those assumptions explicitly, and the groundedness detector flags the mismatch.
> We appreciate the reviewer’s recognition of the empirical strength of the work and thank them for the constructive feedback. We will incorporate the clarifications above to better motivate the structure of the paper and improve interpretability of the method’s behavior.

---

> ### Comment · Reviewer_51RG · 2025-11-25
> **Official Comment by Reviewer 51RG**
>
> Thank the authors for the rebuttal. The rebuttal partly addresses my concern, though I still think the paper needs major revision. I will maintain my score.

---

### Official Review · Reviewer_ohYZ · 2025-11-03

**Soundness:** 2
**Presentation:** 2
**Contribution:** 3
**Rating:** 4
**Confidence:** 3

**Summary:**

The paper investigates how reasoning traces in large language models (LLMs) affect their ability to abstain, that is, to decide when not to answer. The authors demonstrate that while Chain-of-Thought (CoT) prompting improves reasoning accuracy, it consistently harms abstention reliability by making models more prone to overconfident or misleading answers. Across eight established benchmarks and five model families, they show that adding CoT reduces both reliable accuracy (accuracy when answering) and abstention accuracy (correctness of abstain or answer decisions). To address this, they introduce Trace Inversion, a new abstention framework that treats hallucination as the model “answering the wrong question.” Instead of relying on confidence scores, Trace Inversion reconstructs the question implicitly answered by the model’s reasoning trace and measures its similarity to the original query; if misaligned, the model should abstain. Three variants of this method are explored: embedding-based, LLM-based, and grounded detection. Experiments show that these methods outperform all existing abstention baselines and improve reliability

**Strengths:**

The paper correctly identifies a key issue overlooked in prior work: improvements in reasoning performance can sometimes come at the cost of reduced abstention reliability. To address this, it introduces an elegant and effective method called Trace Inversion. The approach is conceptually simple yet broadly applicable, demonstrating strong performance across multiple models and benchmarks. Trace Inversion is particularly compelling because it frames model errors as instances of “answering a different question,” offering a meaningful bridge between interpretability and uncertainty estimation.

**Weaknesses:**

- The approach relies heavily on the reconstructed query and on the hypothesis that hallucinations arise when a model answers an incorrect question. However, this assumption may not always hold. The reconstructed query itself can be inaccurate or incoherent, leading to spurious or unjustified abstentions.

- Using semantic similarity to compare the original question
𝑞
 and the reconstructed question
𝑞
′
 is a fragile proxy for alignment. Similarity metrics may conflate paraphrasing with correctness and fail to capture deeper logical or causal mismatches. How is this handled?

- Trace Inversion still depends on the model’s Chain-of-Thought, making it unable to detect or correct flawed reasoning that appears coherent on the surface. If the CoT itself is wrong, inversion merely reflects that error.

- The paper does not analyze how model accuracy changes with or without Chain-of-Thought. Exploring this trade-off could reveal when one might prefer a model that is slightly less accurate but more cautious or better at abstaining.

- All the models experimented upon are trained with instruction tuning to generate a reasoning chain. My assumption is that they may struggle to generate an answer directly. Wouldn't experiments with base models work better?

- The presentation of the results and experimental variants is somewhat difficult to follow. Clearer tables, visualizations, or structured explanations of the baselines and Trace Inversion variants would make the work more accessible.

- Overall this is an interesting paper and I'm happy to reconsider once my concerns have been addressed

**Questions:**

In weaknesses

---

> ### Author Response · Authors · 2025-11-22
> **Response to Reviewer ohYZ**
>
> We sincerely appreciate the reviewer’s thoughtful and constructive feedback on our work. We address the concerns below and will incorporate the clarifications into the final version.
>
> Our approach does not rely on a single proxy for determining similarity between the original query q and the reconstructed query q′. This is precisely why we introduce three complementary comparison methods: (1) sentence-embedding similarity, (2) a groundedness detector, and (3) token-level overlap. While sentence-embedding similarity is indeed too coarse to capture deeper logical or causal mismatches, it performs well in cases where the model reconstructs the user’s intent but adds stylistic or syntactic variation or additional (but benign) details. In contrast, the groundedness detector explicitly evaluates whether the content of q′ is justified by the original query. This detector is designed to catch exactly the kinds of logical or causal divergences that simple similarity metrics miss. As shown in our examples on L344–357, the detector correctly flags reconstructions that introduce assumptions not present in the original query. Token-level comparisons further capture cases where hallucinated entities or constraints appear in q′. We will revise the paper to make the roles of these three methods clearer and to better illustrate how they jointly mitigate the concern that reconstruction errors could lead to unjustified abstentions.
>
> Regarding the use of a model’s Chain-of-Thought in Trace Inversion, we want to emphasize that the goal of our method is not to correct faulty reasoning but to assess self-consistency between the model’s internal reasoning trace and the question it is answering (see lines 073-074). Even if the model produces a coherent but incorrect CoT, the groundedness detector provides an additional safeguard by checking whether the reconstructed reasoning is actually supported by the original query. Thus, incorrect but fluent reasoning does not automatically pass our alignment test. We will clarify this distinction to avoid the impression that Trace Inversion attempts to repair incorrect CoT traces.
>
> The reviewer also asks about the trade-off between accuracy and abstention with or without CoT. Prior work consistently shows that CoT improves task accuracy across a wide range of benchmarks [1,2]. Our focus in this work is on hallucination-aware abstention rather than performance comparison across prompting paradigms, but we agree that directly evaluating accuracy/coverage trade-offs across CoT settings is a valuable extension. We will include this as a future-work discussion.
>
> Concerning model choice, we use instruction-tuned models because they represent the models deployed in real applications, where hallucination detection is most consequential. Base models, in contrast, struggle to follow instructions reliably [3] and often fail to produce reconstructable reasoning traces. In preliminary experiments, base models failed to generate usable CoT outputs in a large fraction of cases. Finally, we appreciate the comment on presentation clarity. In the revision, we will add a consolidated table (see Table 1) summarizing all Trace Inversion variants and baselines, streamline variant descriptions, and include clearer visualizations where appropriate. We agree this will make the work more accessible.
>
> We thank the reviewer again for the constructive feedback. We believe these clarifications and planned revisions will comprehensively address the concerns and significantly strengthen the paper.
>
> Table 1: Aggregated Table
> | Method        | M     | P     | Q     | D     | G     |
> |---------------|-------|-------|-------|-------|-------|
> | TOKENPROB     | 0.560 | 0.403 | 0.616 | 0.580 | 0.584 |
> | ASKCALI       | 0.657 | 0.558 | 0.627 | 0.645 | 0.663 |
> | REFLECT       | 0.573 | 0.508 | 0.596 | 0.570 | 0.561 |
> | COOPERATE     | 0.593 | 0.481 | 0.563 | 0.538 | 0.548 |
> | COMPETE       | 0.605 | 0.618 | 0.668 | 0.629 | 0.607 |
> | SC            | 0.529 | 0.388 | 0.396 | 0.539 | 0.523 |
> | ATC           | 0.604 | 0.503 | 0.486 | 0.501 | 0.589 |
> | **TrInv-SE**      | 0.562 | 0.450 | 0.564 | **0.711** | 0.619 |
> | **TrInv-LLM**     | 0.629 | 0.627 | 0.566 | 0.680 | 0.673 |
> | **TrInv-GROUND**  |**0.675** | **0.647** | **0.673** | 0.625 | **0.703** |
>
> [1] Wei, Jason, et al. "Chain-of-thought prompting elicits reasoning in large language models." *Advances in neural information processing systems* 35 (2022): 24824-24837.
>
> [2] Kojima, Takeshi, et al. "Large language models are zero-shot reasoners." *Advances in neural information processing systems* 35 (2022): 22199-22213.
>
> [3] Verma, Pulkit, et al. "Teaching LLMs to Plan: Logical Chain-of-Thought Instruction Tuning for Symbolic Planning." *arXiv preprint arXiv:2509.13351* (2025).

---

> ### Author Response · Authors · 2025-11-26
>
> Thank you again for your thoughtful initial feedback. We wanted to kindly check in to see if any additional clarification would be helpful. We are happy to provide further details or address any outstanding concerns.

---

### Note · Authors · 2026-01-02

I have read and agree with the venue's withdrawal policy on behalf of myself and my co-authors.